# Maritime Vessel Classification to Monitor Fisheries with SAR: Demonstration in the North Sea

**Boris Snapir** [1,*] , **Toby W. Waine** [1] and **Lauren Biermann** [2]

[1] School of Water, Energy and Environment, Cranfield University, Cranfield MK43 0AL, UK; t.w.waine@cranfield.ac.uk

[2] Plymouth Marine Laboratory, Prospect Place, Plymouth PL1 3DH, UK; lbi@pml.ac.uk

[*] Correspondence: snapir.boris@gmail.com

**Abstract:** Integration of methods based on satellite remote sensing into current maritime monitoring strategies could help tackle the problem of global overfishing. Operational software is now available to perform vessel *detection* on satellite imagery, but research on vessel *classification* has mainly focused on bulk carriers, container ships, and oil tankers, using high-resolution commercial Synthetic Aperture Radar (SAR) imagery. Here, we present a method based on Random Forest (RF) to distinguish fishing and non-fishing vessels, and apply it to an area in the North Sea. The RF classifier takes as input the vessel's length, longitude, and latitude, its distance to the nearest shore, and the time of the measurement (*am* or *pm*). The classifier is trained and tested on data from the Automatic Identification System (AIS). The overall classification accuracy is 91%, but the precision for the fishing class is only 58% because of specific regions in the study area where activities of fishing and non-fishing vessels overlap. We then apply the classifier to a collection of vessel detections obtained by applying the Search for Unidentified Maritime Objects (SUMO) vessel detector to the 2017 Sentinel-1 SAR images of the North Sea. The trend in our monthly fishing-vessel count agrees with data from Global Fishing Watch on fishing-vessel presence. These initial results suggest that our approach could help monitor intensification or reduction of fishing activity, which is critical in the context of the global overfishing problem.

**Keywords:** fisheries; vessel classification; Sentinel-1; Machine Learning; AIS

## 1. Introduction

According to the Sustainable Development Goal #14 (SDG 14) identified by the United Nations (UN) [1], overfishing is a global challenge which "reduces food production, impairs the functioning of ecosystems and reduces biodiversity". Improving the sustainable use of marine resources requires monitoring tools which can provide long-term observations of (i) the world's fish stocks and (ii) fishing fleets' activity. The latter can now be monitored by several systems. In particular, Automatic Identification System (AIS) and more recently Vessel Monitoring System (VMS) provide huge volumes of data on the geographical position and the identity of vessels around the world. AIS and VMS are cooperative systems that rely on vessels to report their information. These systems are compulsory for vessels of the European Union (EU) which are larger than 15 m [2], and vessel data can be collected from shore-based or satellite receivers.

Satellite Remote Sensing (RS), especially Synthetic Aperture Radar (SAR), can also be used to monitor vessels' position and is complementary to AIS and VMS as it does not rely on vessels' cooperation

[3,4]. Several research groups have developed operational automatic vessel detectors [5]. In parallel, the availability of satellite radar data has also greatly improved with the launch of the Sentinel-1A and Sentinel-1B (S-1) SAR satellites in 2014 and 2016, respectively. This satellite mission, part of the European Space Agency's (ESA) Copernicus program, continuously generates about 6 TB of freely available imagery per day [6]. These free data have already shown value for monitoring marine activities. For example, [7] applied the Search for Unidentified Maritime Objects (SUMO) vessel detector to ∼11,500 S-1A images collected over the Mediterranean Sea between October 2014 and September 2016, and obtained more than 600,000 vessel detections.

Although vessel detection with SAR is already operational, vessel classification (i.e., attributing a vessel type to the detected vessels) remains challenging. This is especially true at the spatial resolution of S-1 (20 m), because it is difficult to extract geometric and radiometric features representative of the different vessel types [8]. Previous research on vessel classification has focused on the use of commercial high-resolution (1–2 m) SAR imagery from TerraSAR-X and COSMO-SkyMed [9] and on three vessel types, namely bulk carriers, container ships, and oil tankers, which represent 70–80% of the vessels worldwide [10]. Several authors used both geometric (length, width, aspect ratio) and radiometric (mean intensity, standard deviation, fractal dimension) features as input of the vessel classifier, and obtained classification accuracy above 90% [11–15]. In particular, in [13] the authors tested their approach on high-resolution (3 m) TerraSAR-X images and medium-resolution (10 m) Radarsat-2 images, and obtained classification accuracy above 90% and below 50%, respectively. They found that for low-resolution images, geometric features played a dominant role in the classification, and radiometric features were only useful at higher resolution, resulting in higher classification accuracy [13,14]. More recently, [16] classified cargo, tanker, windmill, platform, and harbor structure, with an overall accuracy of 94% using a Convolutional Neural Network (CNN). [17] also reported high classification accuracy (98%) with TerraSAR-X images (1 m resolution), using features obtained from Histogram of Oriented Gradients (HOG) combined with Manifold learning for dimensionality reduction, and a Task-Driven Dictionary Learning framework.

To the authors' knowledge, there have not been studies on the classification of SAR detections as fishing or non-fishing vessels. Although fishing vessels are not as numerous as bulk carriers, container ships, and oil tankers, classifying them could support the monitoring of fisheries and help tackle problems around overfishing. In this paper, we present an approach based on Random Forest (RF) trained to classify fishing and non-fishing vessels based on their geographical location (longitude, latitude), an estimate of their length, their distance to the nearest shore, and the time of the measurement (*am* or *pm*). The training phase relies on labelled AIS data. After cross-validation, we apply the RF classifier to a collection of vessel detections obtained by processing S-1 images of the North Sea using the SUMO detector. We show that the variation in fishing-vessel count obtained with our technique agrees with Global Fishing Watch data on fishing-vessel presence. Apart from the initial training phase which relies on commercial AIS data, our approach uses open-source software and free S-1 imagery, making it a cost-effective method to monitor fishing patterns.

## 2. Materials

### 2.1. Vessel Detection with Sentinel-1

The collection of vessel detections used in this study was obtained by applying the SUMO detector to all the 2017 S-1 images acquired over the area shown on Figure 1a. The boundaries of the study area were limited by the extent of the AIS data. On average, every location in the study area is covered by 19 images per month. Figure 1b also shows that the total number of S-1 images available for each month of 2017 increased over time as the recently launched S-1B approached full operational capacity. In this area, the standard acquisition mode is dual-polarization (VV+VH) Interferometric Wide (IW) swath (Table 1).

SUMO is an open-source software developed by the European Commission's Joint Research Center (JRC) based on a Constant False Alarm Rate (CFAR) detector [18]. The SUMO detector was applied to Level-1 Ground Range Detected (GRD) images with the system parameters used in [7]. Specifically, land areas were masked by the OpenStreetMap coastline [19] buffered by 250 m, and the false alarm rate was set to $10^{-7}$, and the detection threshold adjustments for VV images and VH images were set to 10.0 and 2.0, respectively.

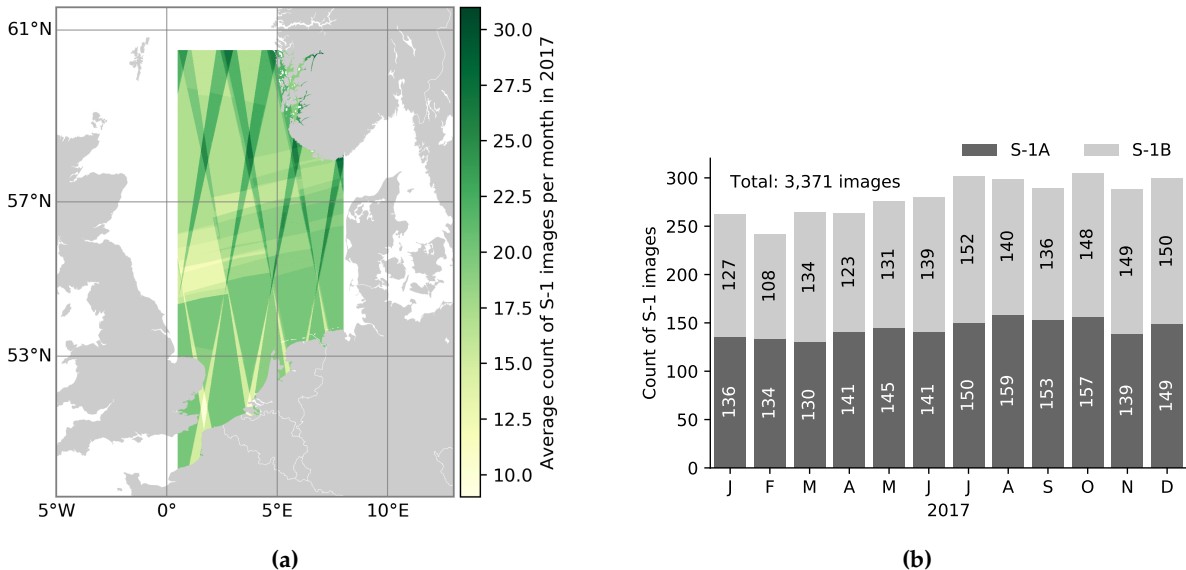

|     |     |
| :-: | :-: |
| (a) | (b) |

**Figure 1.** (**a**) Map of average count of Sentinel-1 images per month for the study area in 2017, and (**b**) monthly count of Sentinel-1A and Sentinel-1B images.

**Table 1.** Technical characteristics of the Sentinel-1 images.

| | |
| ---: | :--- |
| **Acquisition mode** | Interferometric Wide (IW) swath |
| **Processing level** | Level-1 Ground Range Detected (GRD) |
| **Polarization** | VV+VH |
| **Resolution (Range $\times$ Azimuth)** | $20 \times 22$ m |
| **Swath width** | 250 km |
| **Wave length** | C-band |
| **Near field incidence** | 29 deg |
| **Far field incidence** | 46 deg |
| **Local time** | 05:50 or 17:50 |

A total of 868,237 vessels were detected over the 3371 S-1 images. This corresponds to all the VV and VH detections marked as *reliable* by the SUMO software. Figure 2 shows that the vessel density is particularly high (>1000 vessels/10 km$^2$) in the Strait of Dover.

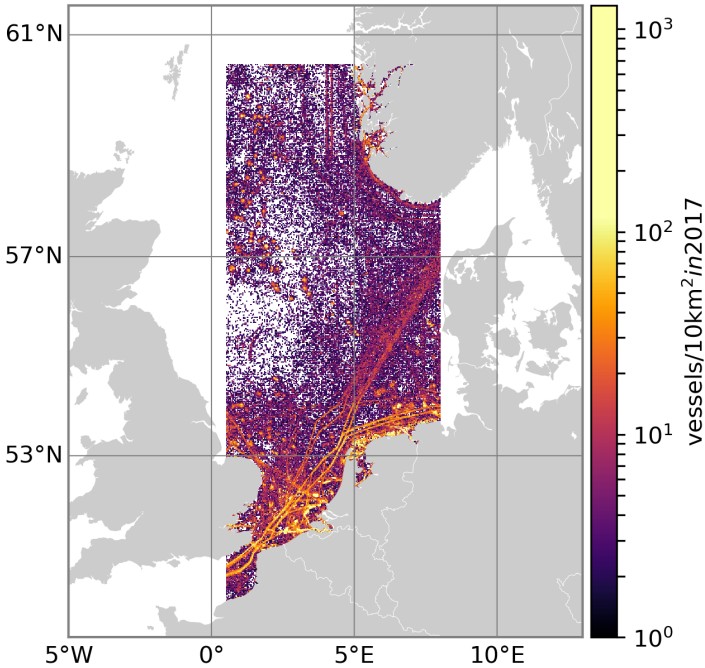

**Figure 2.** Density map of the 868,237 vessels detected by SUMO in 2017.

### 2.2. The Automatic Identification System Data

The AIS data cover the first three months of 2017. This dataset is from AIS messages recorded via satellites. The sampling frequency of the recording is about one hour for each vessel. Figure 3 shows that there is a total of more than 1 million AIS samples, with fishing vessels only accounting for a small portion (12.6%) of the AIS data in the study area. Figure 4 shows the spatial distribution of the fishing and non-fishing vessels, which forms the basis of our approach. Overall, fishing vessels tend to operate in areas away from other vessels. The geographical extent of the AIS data is larger than that of the S-1 data (Figure 2) to avoid any edge effects. In addition to the vessel's geographical position and its type, AIS messages also include information such as length, width, satellite time stamp, Maritime Mobile Service Identity, flag, etc. As explained in the next section, our approach relies only on AIS information which is also available from the SUMO detector.

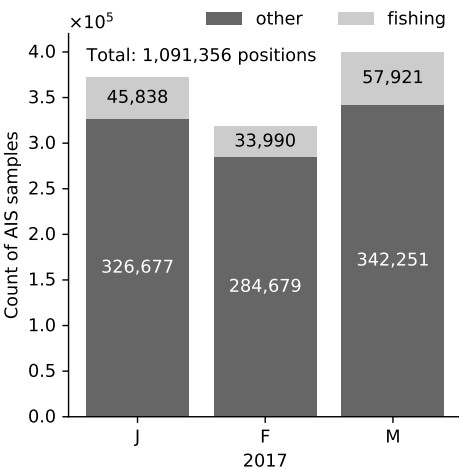

**Figure 3.** Monthly count of AIS samples for fishing and non-fishing (other) vessels.

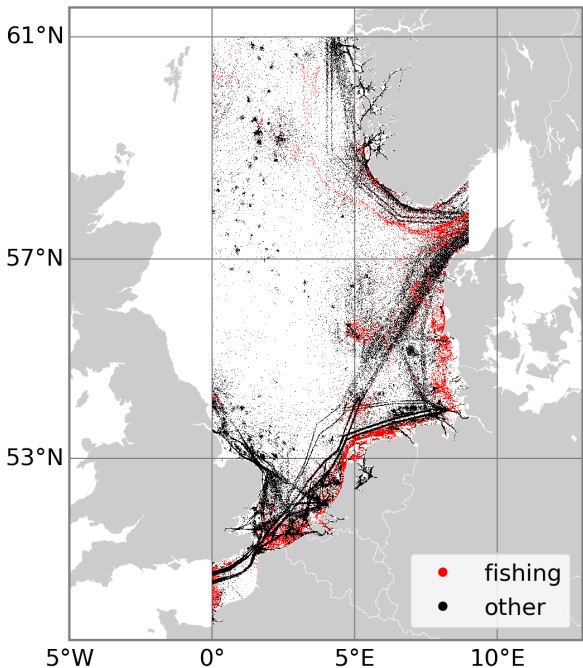

**Figure 4.** Distribution of AIS samples for fishing vessels (red) and non-fishing (other) vessels (black), for the period January–March 2017.

## 3. Methods

Figure 5 summarizes our two-step approach. First, the RF classifier is trained and tested on the AIS dataset. The classifier differentiates fishing and non-fishing vessels based on their longitude, latitude, length, distance to nearest shore, and the time of the measurement. Once trained and tested, the RF is applied to the S-1 detections returned by SUMO to isolate the fishing vessels.

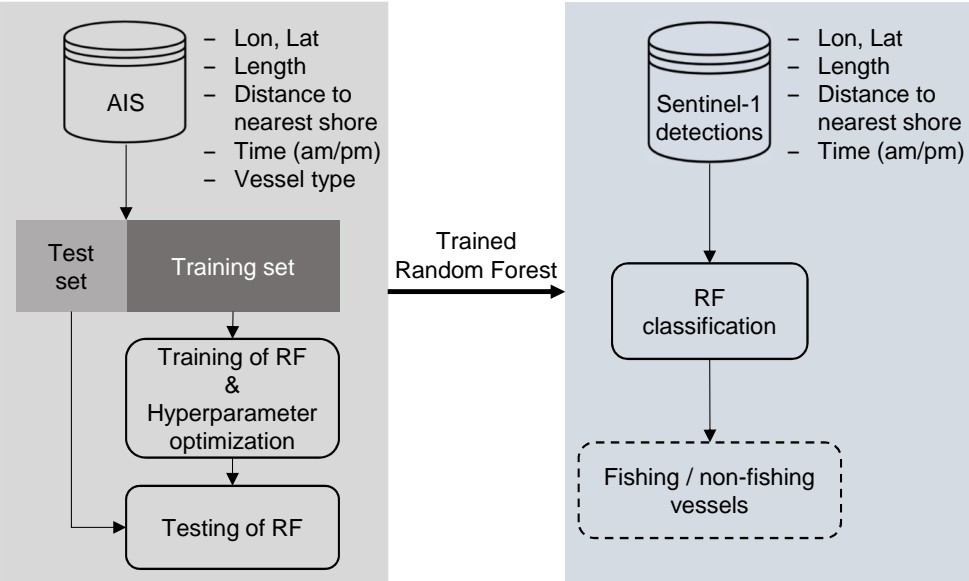

**Figure 5.** The Random Forest classifier is trained and tested on the AIS dataset to classify fishing and non-fishing vessels based on their geographical position (longitude, latitude), their length, their distance to the nearest shore, and whether it is a morning or afternoon measurement. After training and testing, the classifier is applied to the Sentinel-1 detections to isolate the fishing vessels.

*3.1. Random Forest Classifier*

RF is a non-linear supervised classification technique which corresponds to an ensemble of decision trees trained on various subsets of the training set [20,21]. Because single decision trees tend to overfit the data, the underlying idea of RF is that averaging the output of multiple overfitting decision trees trained on random subsets of the dataset, may reduce the overfitting problem and improve the classifier accuracy when applied to new data. While other non-linear classifiers such as Neural Networks are often criticized for being *black boxes* (the classification rules cannot easily be explained), RF can be considered as a *grey box* [22]. In particular, it allows exploring the relative importance of the different input features. Briefly, a decision tree operates as follows. Each node of the tree splits the input data into 2 subsets based on a logical if-then condition on one input feature [23]. The feature and its splitting condition are chosen so that they minimize the impurity of the 2 subsets. This partitioning is repeated until there is less than a given number of data samples at each final node, or until the maximum depth of tree has been reached. For fully grown trees, the computational time of this learning algorithm increases as $\mathcal{O}(n_{\text{tree}} n_{\text{feat}} \log n_{\text{training}})$, with $n_{\text{tree}}$ the number of trees in the forest, $n_{\text{feat}}$ the number of input features considered to choose the splits, and $n_{\text{training}}$ the number of training samples [23]. After training, the predicted output at each final node corresponds to the majority class in that node.

To create the training set, we randomly selected $n$ samples of fishing vessels and $n$ samples of non-fishing vessels. $n$ was set to 96,424 which corresponds to 70% of the fishing-vessel samples. Having the same number of fishing and non-fishing samples ensures that the RF is not biased toward the dominant class (non-fishing vessels). The RF hyperparameters are set via a randomized search with 50 iterations. During the search, each combination of hyperparameters is tested by computing the average classification accuracy over a 5-fold cross-validation. Table 2 summarizes the possible values for the different hyperparameters of the randomized search, other hyperparameters are left at their default value. After 50 iterations, the best combination of hyperparameters is selected and the RF is retrained on the whole training set. The average accuracy of the trained RF is then assessed on 50 test sets. Each test set is

created by randomly sampling with replacement the fishing and non-fishing samples from the left-out samples (41,325 fishing samples, 857,183 non-fishing samples). The number of samples is chosen to match the sample proportion of each class (12.6% fishing, 87.4% non-fishing; Figure 3), i.e., 41,325 fishing samples and 286,084 non-fishing samples in each test sets.

**Table 2.** List of the hyperparameters and their possible values used in the randomized search. $\mathcal{U}\{a, b\}$ is the discrete uniform distribution between $a$ and $b$. When the maximum tree depth is set to *None*, the nodes are expanded until all leaves are pure.

| Hyperparameters | Values |
| --- | --- |
| Number of trees | $\mathcal{U}\{200, 500\}$ |
| min. samples per leaf | $\mathcal{U}\{1, 100\}$ |
| max. tree depth | *None*, 10, 20, 30, 40, 50 |

### 3.2. Input Features

The input features must be available both in the AIS data and the S-1 data. The longitude and latitude are directly available in both datasets. For each AIS sample and S-1 sample, we computed the distance to nearest shore using a 1 km resolution land mask created from the OpenStreetMap coastline. The underlying idea is that fishing vessels seem to operate in areas which are at specific range to the nearest land (see Figure 4). Figure 6a shows the distribution of this feature for the fishing and non-fishing vessels. The significant overlap between the two distributions indicates that this feature alone is not able to accurately discriminate the two vessel classes. As mentioned in the introduction, previous research reported that the vessel length is a geometric feature which is useful to discriminate vessel types with 10-m resolution SAR images [13,14]. Figure 6b confirms that the vessel length will be an important feature in the vessel classification. However, the length estimate returned by SUMO may not be as accurate as the AIS length, especially for small vessels because of the limited image resolution—an average underestimation of –28.83 m was reported in [5] using COSMO-SkyMed and Radarsat-2 images (5–25 m resolution). To estimate the length, SUMO uses a series of thresholds to isolate the pixels which are part of the outline of the vessel [18]. The pixels are then grouped together based on 8-point connectivity and a least-squares line is fitted to the cluster of pixels. The vessel length is taken as the distance between the extreme pixels along the fitted line. Because of the potential mismatch between the AIS length and the SUMO length, the classification rules which may be learned from the AIS length may not be applicable to the length estimated by SUMO. To make the two datasets more comparable, we simplified the length into a categorical feature which can only take three values: *small* (length < 50), *medium* ($50 \leqslant$ length < 100), or *large* (length > 100). Similarly, the time of the day of a sample is also a feature which is different between the AIS data and the S-1 data. The S-1 samples are measured either around 05:50 (descending pass) or 17:50 (ascending pass), while the AIS samples are measured at any time of the day. Although these is a significant overlap between the two vessel classes, Figure 6c shows that fishing vessels tend to operate more in the morning (02:00–11:00), while other vessels operate more uniformly throughout the day. To make the time feature from the AIS and from S-1 comparable, we used only the AIS samples between 03:00 and 09:00 and between 15:00 and 21:00, and marked them as *am* samples and *pm* samples, respectively. The time intervals are centered on the S-1 acquisition times and were chosen to retain enough samples for training/testing of the RF.

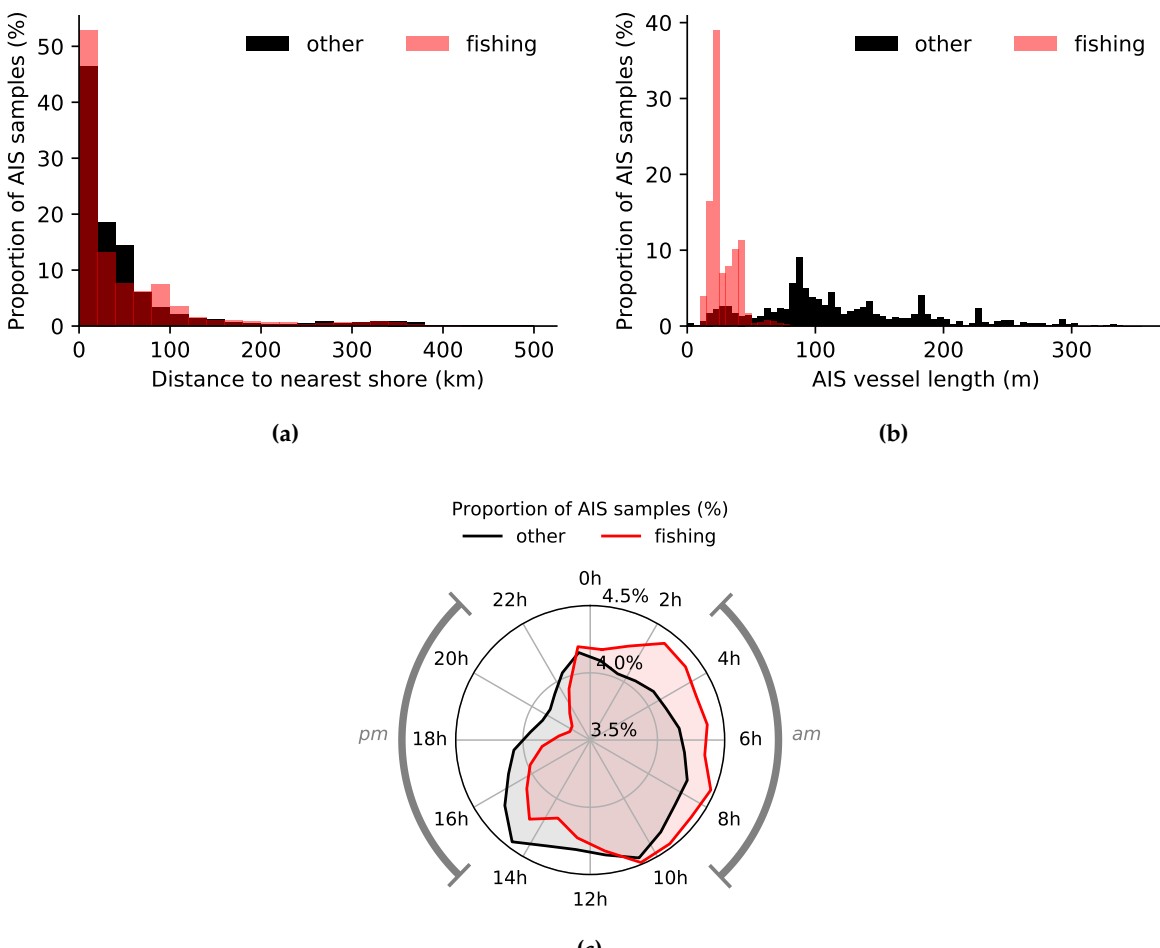

**Figure 6.** Distribution of (**a**) the distance to nearest shore, (**b**) the length, and (**c**) the time of the day for all the fishing (red) and non-fishing (other; black) samples of the AIS data. For (**c**), only the samples within the *am* and *pm* segments are used for training. The total number of fishing and non-fishing samples are 137,749 and 953,607, respectively.

## 3.3. Sentinel-1 Vessel Count

Once the RF classifier has been trained and tested on the AIS data, it is applied to the S-1 detections, and monthly statistics can be computed, for example a monthly count of fishing vessels. The statistics must account for the variation in the number of S-1 images available each month (see Figure 1). In practice, we split the study area into 100×100 km cells (Figure 7a), and for each cell we remove the linear trend between monthly vessel count and monthly S-1 image count (Figure 7b). The cell size is a trade-off between (i) having a reliable count of S-1 detections, and having a correction local enough to capture the spatial heterogeneity in number of S-1 images and in the vessel count. The ratio between corrected count and non-corrected count is then used as a correction factor when computing the monthly counts of fishing and non-fishing vessels.

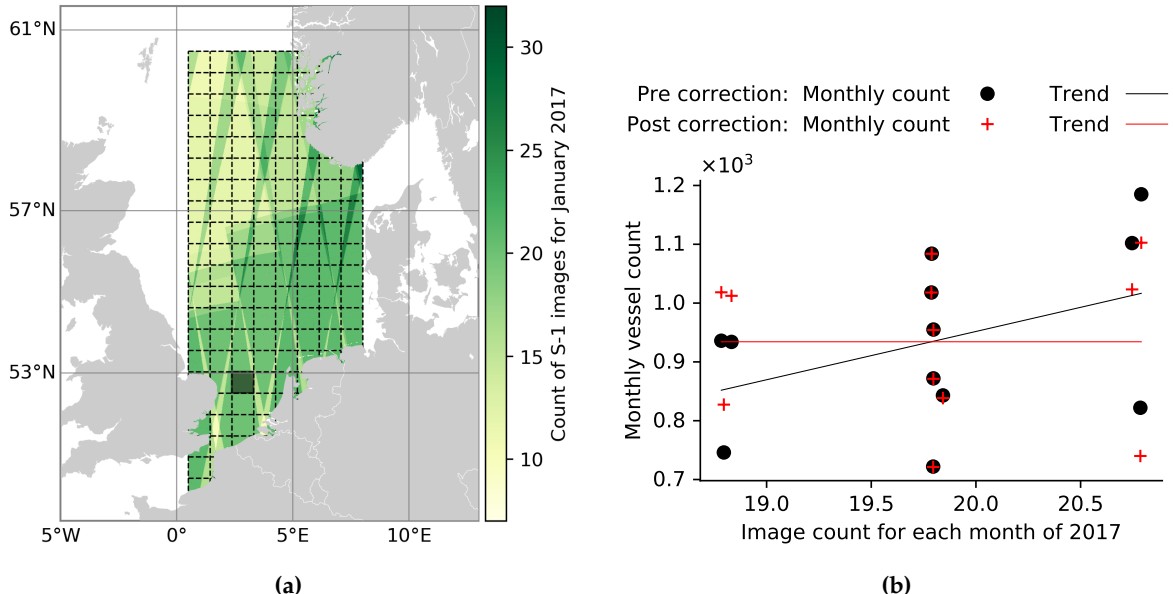

**(a)**                                                                     **(b)**

**Figure 7.** (**a**) The linear trend between monthly vessel count and monthly Sentinel-1 image count is computed for each cell of the grid (dashed lines). The correction for the shaded cell is shown in (**b**), it consists in removing the positive linear trend between monthly vessel count and monthly S-1 image count.

## 4. Results

### 4.1. Training and Testing of Random Forest with the AIS Data

The training phase takes about 30 minutes on a standard computer (Intel Core i7-8650U CPU, 8GB of RAM). The randomized search returned the optimal set of hyperparameters summarized in Table 3, with an average classification accuracy of 0.92 over the 5-fold cross-validation.

**Table 3.** Optimal set of hyperparameters returned by the randomized search.

| Hyperparameter | Value |
| --- | --- |
| Number of trees | 319 |
| min. samples per leaf | 4 |
| max. tree depth | 20 |

Table 4 shows the average confusion matrix when the trained RF is applied to the 50 test sets. Many non-fishing samples (28,613), compared to the total number of true fishing samples (41,325), are wrongly classified as fishing samples. This is mainly because the non-fishing class is much more represented (x7) than the fishing class, and is clearly highlighted by the low classification precision (0.58) for the fishing class in Table 4b. Although the overall accuracy is 0.91, the number of fishing vessels based on our classification is over-estimated by 64%. If the numbers of fishing and non-fishing samples were similar, the performance of the classifier would be good for both classes. This is confirmed by Table 5 which shows the performance assessment over 50 test sets with equal numbers (41,325) of fishing and non-fishing samples. This time, the precision and recall are above 0.9 for both classes, which confirms that the classifier is equally good at classifying fishing and non-fishing vessels.

**Table 4.** Accuracy assessment using test sets composed of 12.6% of fishing samples and 87.4% of non-fishing samples. (**a**) Average confusion matrix over the 50 test sets and (**b**) corresponding classification metrics.

(a)

| Predicted | Truth | |
|---|---|---|
| | **Fishing** | **Other** |
| **Fishing** | 38,979 | 28,613 |
| **Other** | 2346 | 257,471 |

(b)

| | Fishing | Other |
|---|---|---|
| **Precision** | 0.58 | 0.99 |
| **Recall** | 0.94 | 0.90 |
| **F1 score** | 0.72 | 0.94 |
| **Accuracy** | 0.91 | |
| **Kappa coefficient** | 0.66 | |

**Table 5.** Accuracy assessment using balanced test sets (equal numbers of fishing and non-fishing samples). (**a**) Average confusion matrix over the 50 test sets and (**b**) corresponding classification metrics.

(a)

| Predicted | Truth | |
|---|---|---|
| | **Fishing** | **Other** |
| **Fishing** | 39,058 | 4239 |
| **Other** | 2265 | 37,086 |

(b)

| | Fishing | Other |
|---|---|---|
| **Precision** | 0.90 | 0.94 |
| **Recall** | 0.95 | 0.90 |
| **F1 score** | 0.92 | 0.92 |
| **Accuracy** | 0.92 | |
| **Kappa coefficient** | 0.84 | |

Further to this, Figure 8 shows that there are regions (rectangle A) of the study area where the classification precision for the fishing class is high ($\geqslant 0.8$). Low precision (rectangle B) occurs in areas where the non-fishing class dominates, and where the input features may not enable the discrimination of the two classes (fishing and non-fishing vessels in similar location, with similar length, etc.). It follows that special care must be taken if the classifier is applied to a single vessel at a given location. The chance of the classifier to be right about that single vessel is reflected by the performance of the classifier at that specific location.

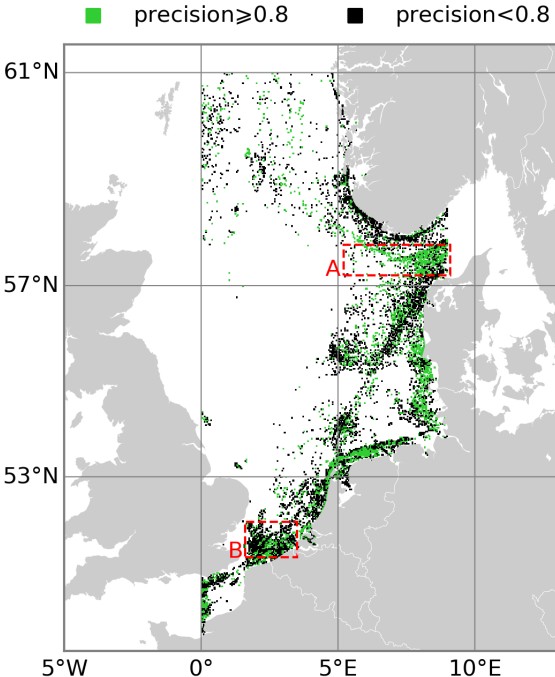

**Figure 8.** Classification precision for the fishing class. The precision is evaluated over 5 × 5 km bins, for a single test set. Rectangle A and rectangle B highlight zones of high precision and low precision, respectively.

Figure 9 shows that the vessel length is the most important input feature in the classification, even after simplifying the length information into the small, medium, and large categories. The latitude is more important than the longitude because the study area spans more latitudes than longitudes, thus latitude discriminates more vessels. The distance to nearest shore is also useful in the classification. In comparison, the am/pm feature has a negligible importance and could be removed from the classifier.

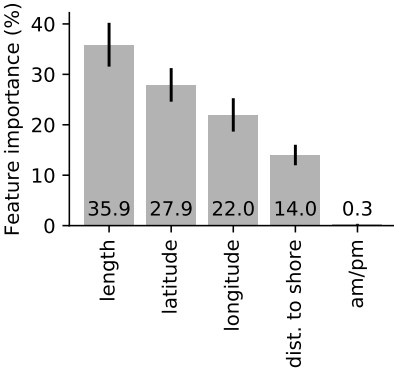

**Figure 9.** Feature importance of the trained Random Forest. The error bar is the standard deviation of the feature importance over the ensemble of decision trees.

*4.2. Trained Random Forest Applied to Sentinel-1 Data*

Figure 10 shows the output of the classifier applied to the S-1 data (144,744 fishing vessels and 723,493 non-fishing vessels). Overall, the fishing (red) and non-fishing (black) patterns are similar to those observed on Figure 4. SUMO may return false detections which explains why Figure 10 is noisier.

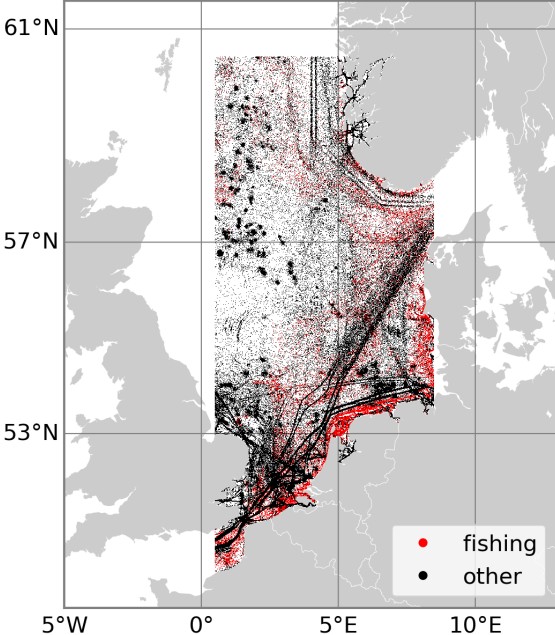

**Figure 10.** Classification of the Sentinel-1 dataset (144,744 fishing vessels and 723,493 non-fishing vessels in 2017).

We then compute the monthly vessel count (Figure 11), with correction presented in Section 3.3. The total vessel count (Figure 11a) is overall constant throughout the year with exception of January and December where there are fewer vessels. The fishing-vessel count (Figure 11b) gradually increases from January to September, then decreases from September to December. Finally, the non-fishing-vessel count (Figure 11c) increases between January and February, then gradually decreases from February to December. From the confusion matrix in Table 4a, one may expect the temporal variation in non-fishing vessels to be similar to that of the fishing vessels because of many samples being wrongly classified as fishing vessels. In practice, the variation in the two vessel classes are significantly different, possibly because the number of samples wrongly classified as fishing vessels remains stable over time.

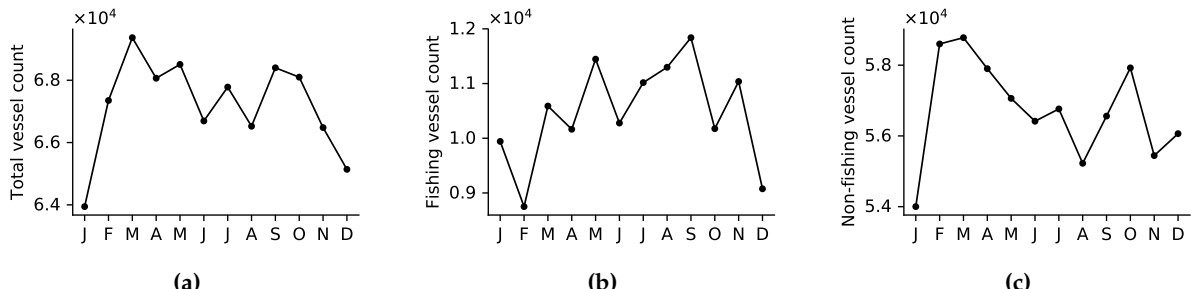

**Figure 11.** Vessel count (2017) after correction for variation in the number of Sentinel-1 images for the (**a**) all the vessels, (**b**) the fishing vessels, and (**c**) the non-fishing vessels in the study area.

To go further, we compare our fishing-vessel count to Global Fishing Watch (GFW) data on *fishing-vessel presence*, available daily from January 2012 to January 2017 at 0.01 deg resolution [24]. GFW identified 6 fishing and 6 non-fishing-vessel classes with an accuracy of 95%, by applying a CNN classifier

to vessel trajectories extracted from AIS data (see [25] for more details). We summed the vessel presence across the fishing classes, then computed the monthly mean over our study area for 2016. Although our data and the GFW data are for different years, there is a reasonable agreement between our fishing-vessel count and the mean fishing-vessel presence (Figure 12a). Both data suggest a peak in fishing-vessel activity around September, while December, January, and February are the less active months. In Figure 12b, we restrict the comparison to the area where the classification precision for the fishing class is above 0.8 (green area in Figure 8). The agreement between the two datasets does not improve, possibly because there are too few samples to perform a reliable comparison.

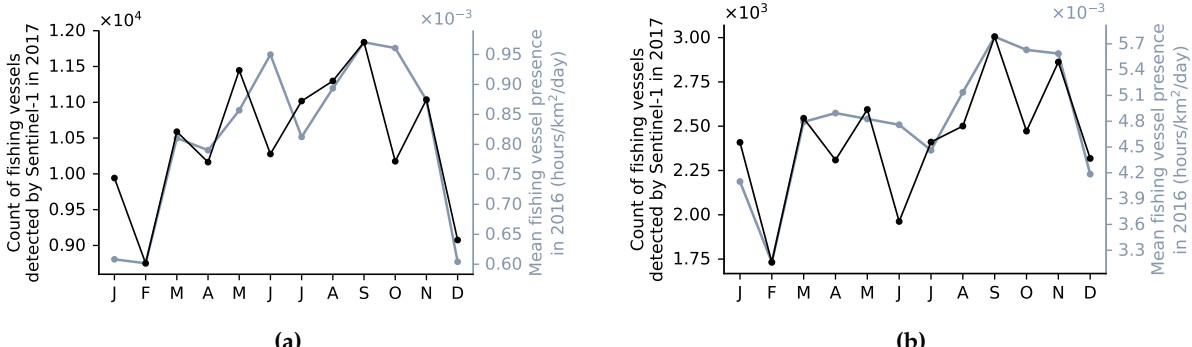

**Figure 12.** Comparison between the count of fishing vessels detected by Sentinel-1 in 2017 and the fishing-vessel presence from the 2016 GFW data (**a**) for the whole study area and (**b**) for the area where the classification precision of the fishing class is above 0.8.

## 5. Discussion

### 5.1. Vessel Detection with SUMO and Sentinel-1

The use of SUMO and S-1 as a monitoring system, although cost-effective, has several limitations. First, S-1 only provides a snapshot of the maritime traffic on average every 1–2 days either at 06:00 or at 18:00 (which is far from the sampling density of the AIS data), and small vessels may not be detectable at the 20 m resolution [26–28].

Second, the SUMO detector includes steps to reject azimuth ambiguities and detections with unrealistic size, but the final output may still contain false detections. These are in part due to features such as strong sea clutter, wave crests, range ambiguities, or unmasked land [5,7,18]. Systematic azimuth ambiguities caused by strong land-based scatters could be further removed before applying the classifier, by implementing multi-temporal methods as suggested in [7,29]. Radio Frequency Interference (RFI) [7,30] is another potential source of (non-systematic) false detections. In [7], heavily RFI-affected areas (lines of false detections) were manually removed from the analysis. Although no obvious RFI were observed in our study area, some RFI may have affected our final vessel counts. As a result, our vessel count estimate should not be interpreted as an absolute count of vessels. Despite these limitations, our results show that the spatial and temporal variations in this estimate likely reflect true variations in the actual number of vessels.

Finally, relatively high detection thresholds have been selected to limit the amount of false detections as done in [7]. Consequently, the smaller vessels may not be detected by SUMO. This is an important limitation since fishing vessels can be small (Figure 6b), hence our vessel count may only reflect a portion of all the fishing vessels.

*5.2. Input Features of the Classifier*

5.2.1. Other Input Features

The feature importance analysis confirmed that the vessel length is critical at the S-1 resolution as found in [13,14]. Although radiometric features were found to be useful for high-resolution imagery alone, previous research only focused on classifying bulk carriers, container ships, and oil tankers. Future research could assess if radiometric features may help in discriminating fishing and non-fishing vessels. Nonetheless, radiometric features have several disadvantages. First, S-1 detections must be matched to AIS samples to create a labelled training set of S-1 detections as done in [8]—this requires interpolating AIS vessel trajectories at the time of the S-1 acquisitions and may lead to erroneous matching [7,8]. Second, radiometric features are sensor specific, for example they are different at C-band and X-band. In comparison, our current method is sensor agnostic—it can be used with both SAR and optical instruments as long as estimates of the vessels' position and length are available.

Since both our vessel count and the GFW data show monthly variations in fishing activity, the month of the year may be a helpful input feature to classify fishing vessels. This could be tested with AIS data spanning a full year.

5.2.2. Generalization of the Classification

The latitude and longitude were determinant in our classification, but they inherently reduce the generalization capability of the classifier. First, the classification rules associated with these two features are specific to the study area. Applying the classifier to a different location would require retraining the classifier with AIS data for this new location. Second, our classifier can highlight increase/decrease in fishing vessels in a known fishing zone, but it will *a priori* not cope with fishing vessels operating in a new area not captured by the training set. So, the classifier would need to be retrained over time to learn about spatial changes in fishing zones.

Regarding the monitoring of illegal, unregulated and unreported (IUU) fishing, S-1 can detect vessels which do not carry AIS or which temper with their AIS. The RF classifier will be able to label them as fishing vessels only if they operate in fishing zones captured by the AIS data used for training. In other words, our classifier is best applied in areas used or traversed by fishing vessels with AIS.

Although we focused on classifying fishing vessels, our approach could be adapted to distinguish other vessel types (cargo, tanker, ...), if these vessels operate in different areas and/or have different lengths.

*5.3. Classification Accuracy*

Our results showed that the classification accuracy (especially the precision, Figure 8) changes with the location. In particular, when fishing and non-fishing vessels operate in the same area, the latitude and longitude features may not discriminate the two classes and classification errors may occur. Our test area has regions such as the Strait of Dover (rectangle B on Figure 8), with dense maritime traffic from various vessel types. This makes the classification task particularly difficult. The classifier may perform better for ocean fisheries where there are potentially fewer vessel types and less overlap between their operating zones, except for transshipments [31].

**6. Conclusions**

We presented a method for the classification of SAR vessel detections into fishing and non-fishing-vessel classes. The method relies on a RF classifier with 5 input features: longitude, latitude, length, distance to nearest shore, and time of measurement (am or pm). The classifier was trained and tested on labelled AIS data, and was then applied on S-1 detection provided by the SUMO detector. The

length, latitude, longitude, and distance to nearest shore are the only important features in the classification. When tested on datasets with realistic sample proportions (12.6% fishing, 87.4% non-fishing), the overall classification accuracy is 91%, but the precision of the fishing class is only 58%. Precision, however, varies with the location and is only low in areas where the two vessel classes overlap, and where the non-fishing class dominates. When applied to the 2017 S-1 data, there is a good agreement between our monthly fishing-vessel count and data from GFW on fishing-vessel presence in 2016. Although our vessel count is not a reliable estimate of the actual number of fishing/non-fishing vessels, the results suggest that our method still captures the correct temporal variation in fishing activity. In particular, our approach can be used to indicate intensification or reduction of fishing effort in a given area, which is critical in the context of the global overfishing problem.

**Author Contributions:** Formal analysis, B.S.; Funding acquisition, B.S. and L.B.; Methodology, B.S. and L.B.; Visualization, B.S.; Writing—review & editing, B.S., T.W.W. and L.B.

**Funding:** This research was funded by the Department for Environment, Food & Rural Affairs (DEFRA) Earth Observation Center of Excellence.

**Acknowledgments:** The AIS data were purchased from exactEarth and preprocessed by Inkblot Software.

**Conflicts of Interest:** The authors declare no conflict of interest. The funders had no role in the design of the study; in the collection, analyses, or interpretation of data; in the writing of the manuscript, or in the decision to publish the results.

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
