# Peer review of "Maritime Vessel Classification to Monitor Fisheries with SAR: Demonstration in the North Sea"

_remotesensing, doi:10.3390/rs11030353_

Round 1

Reviewer 1 Report

The new perspective of the paper permits having a new dimension of the problem presented in the first version of the paper. The paper introduces some additional information that permits having a blurred vision of your algorithm complexity. E.g., in section 3.1 the paper reports “[...] non-linear supervised [...]”: this sentence is not enough to define the complexity.

Section 4.1 reports “8Go”, do you mean GB? Moreover, during the 30 minutes, what is the CPU time dedicated to computing the described algorithm? And the memory allocation? Do your algorithm read/writes data to/from some secondary memory? Which one? Thank you for your work, please resolve the problems I put in evidence.

Author Response

Please see attached pdf file.

Reviewer 2 Report

The authors have solved most of my questions. This paper can be published now in principal. However, the methodology mainly depends on the location and the length. Thus, the authors should clarify that if there is a specified other-type ship in the specified area, it may be taken as a fishing ship. This method may be not reliable for identifying a specified ship, but it may be useful for identifying a large number of ships for the over-fishing problem.

Besides, a classical problem in SAR ship detection is the azimuth ambiguities, especially for littoral areas. Thus, it is interesting to know how to remove the false alarms caused by azimuth ambiguities in SUMO. Is there any influence on your methodology? About azimuth ambiguities, you can refer to the following references. (Please feel free to cite them)

[1] Leng, X.; Ji, K.; Zhou, S.; Zou, H. Azimuth Ambiguities Removal in Littoral Zones Based on Multi-Temporal SAR Images. Remote Sensing. 2017, 9, 866.

[2] Velotto, D.; Soccorsi, M.; Lehner, S. Azimuth ambiguities removal for ship detection using full polarimetric X-Band SAR data. IEEE Trans. Geosci. Remote Sens. 2014, 52, 76–88.

[3] Leng, X.; Ji, K.; Zhou, S.; Xing, X.; Zou, H. An adaptive ship detection scheme for spaceborne SAR imagery. Sensors 2016, 16, 1345.

[4] Greidanus, H.; Santamaria, C. First Analyses of Sentinel-1 Images for Maritime Surveillance. In JRC Science and Policy Reports; Publications Office of the European Union: Luxembourg, 2014.

In addition, since Sentinel-1 (C-band) can be easily effected by Radio Frequency Interference (RFI), the effect caused by RFI in ship detection and classification should also be discussed. About RFI in Sentinel-1 data, you can refer to the following references. (Please feel free to cite them)

[1] The above [4].

[2] http://forum.step.esa.int/t/noise-objects-in-sentinel-1-grd-data/7627.

[3] X. Leng, K. Ji, S. Zhou, and H. Zou, “Azimuth ambiguities removal in littoral zones based on multi-temporal SAR images,” Remote Sens., vol. 9, no. 8, p. 866, 2017.

[4] C. Santamaria, M. Alvarez, H. Greidanus, V. Syrris, P. Soille, and P. Argentieri, “Mass processing of Sentinel-1 Images for marine surveillance,” Remote Sensing, vol. 9, no. 7, 678, 2017.

Author Response

Please see attached pdf file.

Reviewer 3 Report

In this paper the authors propose a methodology to classify fishing vessels from SAR imagery using a random forest classification method based on ship detection from the SUMO software and AIS information. The paper is very well written, concise and clear, the subject is interesting and the outcome of this research could lead to pertinent information for the management and monitoring of fisheries. The changes to the paper helped to improve the understanding of the methodology and the interpretation of the results.

I believe that most concerns have been addressed, however I still have a minor comment regarding the use of the SUMO software. The authors mention the false alarm rate and detection thresholds used for the study are the same as the ones used by Santamaria et al. 2017 (Ref [7]) to simplify the analysis and concentrate on the classification step. One thing that should be noted is that Santamaria et al. raised the detection thresholds to enable the analysis of a large set of images and reduce false alarms but consequently smaller boats were not detected in their study. This can have a significant impact on the current study since most fishing boats will tend to be smaller as demonstrated in Figure 6. This fact will not directly impact the classification algorithm but will definitely impact its usability for real world applications. I believe this point should at least be raised in the discussion and could be the subject of future work.

I also noted a few minor language errors:
Line 163: "Although these is significant overlap... " could be "Although there is a significant overlap..."
Line 188: "... may not enable discriminating the two classes ..." could be " ... may not enable the discrimination of the two classes ..."

Overall I would recommend the publication of the paper after minor revisions.

Author Response

Please see attached pdf file.

Round 2

Reviewer 1 Report

Dear authors, I accept the paper with reserve.

This manuscript is a resubmission of an earlier submission. The following is a list of the peer review reports and author responses from that submission.

Round 1

Reviewer 1 Report

The paper faces an interesting  question.

The detection and classification problem to tackle the over-fishing using low-resolution SAR images

is an open-challenge that requires specific study. The proposed solution faces this problem by

using an RF classifier using longitude, latitude, length, distance to nearest shore, and time of

measurement. The opinion of the review is that this set of features do not help to solve the over-

fishing problem it is useful only to monitor particular areas of the seas. In fact, the classifier uses

the position of the ship to determine its class.

The result section reports that “The training phase takes about 30 minutes on a standard desktop

computer”. What is a standard computer? What kind of CPU are you using? It is a little different to

use an i3 or an i7 or an AMD CPU. How much memory do you have? How much memory do you

use during the training phase? Is the algorithm linear respecting the input data? Is it sublinear or

polynomial? Which is the algorithm complexity? Here the reviewer expects that the paper presents

a deep description of the algorithm you used.

Reviewer 2 Report

This paper presents a very interesting topic. It uses Random Forest to distinguish fishing ships and other ships. The features used are simple features collected from AIS and SAR images. Overall, it is well written and worth publication.

There are some minor comments.

How to measure the length of the ship in the SAR images? Which method do the authors use? Does it bias from the result of AIS?

The time is used as a feature. However, the importance is very small? Is it necessary?

Does this method can be applied to other types of ships? e.g., oil ships.

Reviewer 3 Report

In this paper the authors propose a methodology to classify fishing vessels from SAR imagery using a random forest classification method. In order to achieve this, the images are first processed using the SUMO software to extract the position and size of individual vessels which are compared to AIS information to train a random forest classifier. The paper is very well written, concise and clear, the subject is interesting and the outcome of this research could lead to pertinent information for the management and monitoring of fisheries. I only have some minor concerns with the presentation of the results which are detailed below.

My main concerns are related to the accuracy assessment of the classification. In section 3.1, the authors explain that the sampling method for the training phase took into consideration the class imbalance between fishing and non-fishing vessels by using the same number of samples for both classes. However, for the accuracy assessment in Table 4 the authors decided to keep the proportion between fishing and non-fishing vessels (12.6% vs 87.4%) which in my opinion does not give an accurate picture of the accuracy of the classifier. The Kappa coefficient, which should be added to the accuracy assessment, amounts to around 0,66 when calculated from the data in table 4. This would mean that a significant portion of the classification results would be due to random chance given the strong class imbalance. However, this is not the case since the class imbalance was taken into account in the training phase. I believe that the accuracy assessment of the classifier should use a balanced set of samples to reproduce the conditions used for the training of the classifier. The current accuracy assessment still provides some useful information on how the classifier would perform in "real-world" cases where a strong imbalance appears between classes, but I believe it should be used along with a "balanced" accuracy assessment to give a complete picture and show the accuracy of the classifier if it was used in a region where the ratio of vessel types might be different.

The other concern I have is with the use of the SUMO software. The authors mention that there are some problems with the accuracy of the software, but do no provide any detailed analysis. The false detection rate seems really high when looking at Figure 10 so I believe that an independent analysis of SUMO's performance in terms of detection/non-detection of vessels should be added to have an idea of the source of the observed inaccuracies. The authors also mention the parameters used for ship detection in SUMO, which were taken from a previous study, but do not mention if a proper calibration was made to ensure that these are indeed the optimal parameters to use in the current case.

Also, on the use of the radiometric features in section 5.2.1 the authors mention that matching S-1 radiometry with AIS samples requires interpolating AIS trajectory which can lead to erroneous matching. However, this problem would still affect the SUMO software in its current ship detection algorithm in my opinion, which might explain some inaccuracies mentioned in my previous point. The current algorithm already relies heavily on the matching of AIS data with ships detected in SUMO so I don't see how this would be different for radiometric features. The other reasons given in section 5.2.1 are however correct in my opinion, radiometry would not be a reliable feature to use in the RF algorithm if the methodology aims to remain sensor agnostic. One would then only need to calibrate SUMO for different sensors instead of retraining the RF for every sensor, frequency, incidence angle etc.

On the question of features to include in the RF algorithm, the authors used the AM/PM feature and showed that it had negligible importance. However, the authors also show that there are varying temporal trends between fishing and non-fishing vessels. Could the date or season be considered as an input feature then since there seems to be a good class separation for this feature?  

I also have a few minor comments on specific parts of the paper listed below:

On line 119 the authors mention that 337,305 non-fishing samples were used but on line 121 they mention 103,226 non-fishing samples. It is not clear where the difference comes from since there is no difference in the number of fishing samples.

On line 192, it would be useful to add the classification accuracy achieved by GFW with the CNN classifier. I believe the information is useful and simple enough to include in this paper instead of having to consult the original paper.

On line 218, the word "nonetheless" is repeated twice in the same sentence.

Overall I believe this is an interesting and well written paper and should be considered for publication after some minor revisions.